# Double or Nothing: Multiplicative Incentive Mechanisms for Crowdsourcing

**Nihar B. Shah**
University of California, Berkeley
nihar@eecs.berkeley.edu

**Dengyong Zhou**
Microsoft Research
dengyong.zhou@microsoft.com

## Abstract

Crowdsourcing has gained immense popularity in machine learning applications for obtaining large amounts of labeled data. Crowdsourcing is cheap and fast, but suffers from the problem of low-quality data. To address this fundamental challenge in crowdsourcing, we propose a simple payment mechanism to incentivize workers to answer only the questions that they are sure of and skip the rest. We show that surprisingly, under a mild and natural "no-free-lunch" requirement, this mechanism is the one and only incentive-compatible payment mechanism possible. We also show that among all possible incentive-compatible mechanisms (that may or may not satisfy no-free-lunch), our mechanism makes the smallest possible payment to spammers. Interestingly, this unique mechanism takes a "multiplicative" form. The simplicity of the mechanism is an added benefit. In preliminary experiments involving over several hundred workers, we observe a significant reduction in the error rates under our unique mechanism for the same or lower monetary expenditure.

## 1   Introduction

Complex machine learning tools such as deep learning are gaining increasing popularity and are being applied to a wide variety of problems. These tools, however, require large amounts of labeled data [HDY+12, RYZ+10, DDS+09, CBW+10]. These large labeling tasks are being performed by coordinating crowds of semi-skilled workers through the Internet. This is known as crowdsourcing. Crowdsourcing as a means of collecting labeled training data has now become indispensable to the engineering of intelligent systems.

Most workers in crowdsourcing are not experts. As a consequence, labels obtained from crowdsourcing typically have a significant amount of error [KKKMF11, VdVE11, WLC+10]. Recent efforts have focused on developing statistical techniques to post-process the noisy labels in order to improve its quality (e.g., [RYZ+10, ZLP+15, KOS11, IPSW14]). However, when the inputs to these algorithms are erroneous, it is difficult to guarantee that the processed labels will be reliable enough for subsequent use by machine learning or other applications. In order to avoid "garbage in, garbage out", we take a complementary approach to this problem: cleaning the data at the time of collection.

We consider crowdsourcing settings where the workers are paid for their services, such as in the popular crowdsourcing platforms of Amazon Mechanical Turk and others. These commercial platforms have gained substantial popularity due to their support for a diverse range of tasks for machine learning labeling, varying from image annotation and text recognition to speech captioning and machine translation. We consider problems that are objective in nature, that is, have a definite answer. Figure 1a depicts an example of such a question where the worker is shown a set of images, and for each image, the worker is required to identify if the image depicts the Golden Gate Bridge.

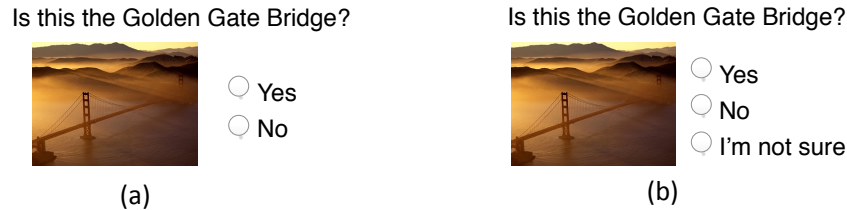

Figure 1: Different interfaces in a crowdsourcing setup: (a) the conventional interface, and (b) with an option to skip.

Our approach builds on the simple insight that in typical crowdsourcing setups, workers are simply paid in proportion to the amount of tasks they complete. As a result, workers attempt to answer questions that they are not sure of, thereby increasing the error rate of the labels. For the questions that a worker is not sure of, her answers could be very unreliable [WLC+10, KKKMF11, VdVE11, JSV14]. To ensure acquisition of only high-quality labels, we wish to encourage the worker to skip the questions about which she is unsure, for instance, by providing an explicit "I'm not sure" option for every question (see Figure 1b). Our goal is to develop payment mechanisms to encourage the worker to select this option when she is unsure. We will term any payment mechanism that incentivizes the worker to do so as "incentive compatible".

In addition to incentive compatibility, preventing spammers is another desirable requirement from incentive mechanisms in crowdsourcing. Spammers are workers who answer randomly without regard to the question being asked, in the hope of earning some free money, and are known to exist in large numbers on crowdsourcing platforms [WLC+10, Boh11, KKKMF11, VdVE11]. It is thus of interest to deter spammers by paying them as low as possible. An intuitive objective, to this end, is to ensure a zero expenditure on spammers who answer randomly. In this paper, however, we impose a strictly and significantly weaker condition, and then show that there is one and only one incentive-compatible mechanism that can satisfy this weak condition. Our requirement, referred to as the "no-free-lunch" axiom, says that if *all* the questions attempted by the worker are answered incorrectly, then the payment must be zero.

We propose a payment mechanism for the aforementioned setting ("incentive compatibility" plus "no-free-lunch"), and show that surprisingly, this is the *only* possible mechanism. We also show that additionally, our mechanism makes the smallest possible payment to spammers among all possible incentive compatible mechanisms that may or may not satisfy the no-free-lunch axiom. Our payment mechanism takes a multiplicative form: the evaluation of the worker's response to each question is a certain score, and the final payment is a *product* of these scores. This mechanism has additional appealing features in that it is simple to compute, and is also simple to explain to the workers. Our mechanism is applicable to any type of objective questions, including multiple choice annotation questions, transcription tasks, etc.

In order to test whether our mechanism is practical, and to assess the quality of the final labels obtained, we conducted experiments on the Amazon Mechanical Turk crowdsourcing platform. In our preliminary experiments that involved over several hundred workers, we found that the quality of data improved by two-fold under our unique mechanism, with the total monetary expenditure being the same or lower as compared to the conventional baseline.

## 2 Problem Setting

In the crowdsourcing setting that we consider, one or more workers perform a *task*, where a task consists of multiple *questions*. The questions are objective, by which we mean, each question has precisely one correct answer. Examples of objective questions include multiple-choice classification questions such as Figure 1, questions on transcribing text from audio or images, etc.

For any possible answer to any question, we define the worker's *confidence about an answer* as the probability, according to her belief, of this answer being correct. In other words, one can assume that the worker has (in her mind) a probability distribution over all possible answers to a question, and the confidence for an answer is the probability of that answer being correct. As a shorthand, we also define the *confidence about a question* as the confidence for the answer that the worker is most

confident about for that question. We assume that the worker's confidences for different questions are independent. Our goal is that for every question, the worker should be incentivized to:

1. skip if the confidence is below a certain pre-defined threshold, otherwise:
2. select the answer that she thinks is most confident about.

More formally, let $T \in (0, 1)$ be a predefined value. The goal is to design payment mechanisms that incentivize the worker to skip the questions for which her confidence is lower than $T$, and attempt those for which her confidence is higher than $T$. [1] Moreover, for the questions that she attempts to answer, she must be incentivized to select the answer that she believes is most likely to be correct. The threshold $T$ may be chosen based on various factors of the problem at hand, for example, on the downstream machine learning algorithms using the crowdsourced data, or the knowledge of the statistics of worker abilities, etc. In this paper we assume that the threshold $T$ is given to us.

Let $N$ denote the total number of questions in the task. Among these, we assume the existence of some "gold standard" questions, that is, a set of questions whose answers are known to the requester. Let $G$ $(1 \leq G \leq N)$ denote the number of gold standard questions. The $G$ gold standard questions are assumed to be distributed uniformly at random in the pool of $N$ questions (of course, the worker does not know which $G$ of the $N$ questions form the gold standard). The payment to a worker for a task is computed after receiving her responses to all the questions in the task. The payment is based on the worker's performance on the gold standard questions. Since the payment is based on known answers, the payments to different workers do not depend on each other, thereby allowing us to consider the presence of only one worker without any loss in generality.

We will employ the following standard notation. For any positive integer $K$, the set $\{1, \dots, K\}$ is denoted by $[K]$. The indicator function is denoted by $\mathbf{1}$, i.e., $\mathbf{1}\{z\} = 1$ if $z$ is true, and $0$ otherwise. The notation $\mathbb{R}_+$ denotes the set of all non-negative real numbers.

Let $x_1, \dots, x_G \in \{-1, 0, +1\}$ denote the evaluations of the answers that the worker gives to the $G$ gold standard questions. Here, "0" denotes that the worker skipped the question, "−1" denotes that the worker attempted to answer the question and that answer was incorrect, and "+1" denotes that the worker attempted to answer the question and that answer was correct. Let $f : \{-1, 0, +1\}^G \rightarrow \mathbb{R}_+$ denote the payment function, namely, a function that determines the payment to the worker based on these evaluations $x_1, \dots, x_G$. Note that the crowdsourcing platforms of today mandate the payments to be non-negative. We will let $\mu$ $(> 0)$ denote the *budget*, i.e., the maximum amount that can be paid to any individual worker for this task:

$$\max_{x_1, \dots, x_G} f(x_1, \dots, x_G) = \mu.$$

The amount $\mu$ is thus the amount of compensation paid to a perfect agent for her work. We will assume this budget condition of $\mu$ throughout the rest of the paper.

We assume that the worker attempts to maximize her overall expected payment. In what follows, the expression 'the worker's expected payment' will refer to the expected payment from the worker's point of view, and the expectation will be taken with respect to the worker's confidences about her answers and the uniformly random choice of the $G$ gold standard questions among the $N$ questions in the task. For any question $i \in [N]$, let $y_i = 1$ if the worker attempts question $i$, and set $y_i = 0$ otherwise. Further, for every question $i \in [N]$ such that $y_i \neq 0$, let $p_i$ be the confidence of the worker for the answer she has selected for question $i$, and for every question $i \in [N]$ such that $y_i = 0$, let $p_i \in (0, 1)$ be any arbitrary value. Let $E = (\epsilon_1, \dots, \epsilon_G) \in \{-1, 1\}^G$. Then from the worker's perspective, the expected payment for the selected answers and confidence-levels is

$$\frac{1}{\binom{N}{G}} \sum_{\substack{(j_1, \dots, j_G) \\ \subseteq \{1, \dots, N\}}} \sum_{E \in \{-1,1\}^G} \left( f(\epsilon_1 y_{j_1}, \dots, \epsilon_G y_{j_G}) \prod_{i=1}^{G} (p_{j_i})^{\frac{1+\epsilon_i}{2}} (1 - p_{j_i})^{\frac{1-\epsilon_i}{2}} \right).$$

In the expression above, the outermost summation corresponds to the expectation with respect to the randomness arising from the unknown choice of the gold standard questions. The inner summation corresponds to the expectation with respect to the worker's beliefs about the correctness of her responses.

We will call any payment function $f$ as an *incentive-compatible mechanism* if the expected payment of the worker under this payment function is *strictly* maximized when the worker responds in the manner desired.[2]

# 3 Main results: Incentive-compatible mechanism and guarantees

In this section, we present the main results of the paper, namely, the design of incentive-compatible mechanisms with practically useful properties. To this end, we impose the following natural requirement on the payment function $f$ that is motivated by the practical considerations of budget constraints and discouraging spammers and miscreants [Boh11, KKKMF11, VdVE11, WLC$^+$10]. We term this requirement as the "no-free-lunch axiom":

**Axiom 1 (No-free-lunch axiom).** *If all the answers attempted by the worker in the gold standard are wrong, then the payment is zero. More formally, for every set of evaluations $(x_1, \ldots, x_G)$ that satisfy $0 < \sum_{i=1}^{G} \mathbf{1}\{x_i \neq 0\} = \sum_{i=1}^{G} \mathbf{1}\{x_i = -1\}$, we require the payment to satisfy $f(x_1, \ldots, x_G) = 0$.*

Observe that no-free-lunch is an extremely mild requirement. In fact, it is significantly weaker than imposing a zero payment on workers who answer randomly. For instance, if the questions are of binary-choice format, then randomly choosing among the two options for each question would result in $50\%$ of the answers being correct in expectation, while the no-free-lunch axiom is applicable only when none of them turns out to be correct.

## 3.1 Proposed "Multiplicative" Mechanism

We now present our proposed payment mechanism in Algorithm 1.

---
**Algorithm 1** "Multiplicative" incentive-compatible mechanism

---
- Inputs: Threshold $T$, Budget $\mu$, Evaluations $(x_1, \ldots, x_G) \in \{-1, 0, +1\}^G$ of the worker's answers to the $G$ gold standard questions
- Let $C = \sum_{i=1}^{G} \mathbf{1}\{x_i = 1\}$ and $W = \sum_{i=1}^{G} \mathbf{1}\{x_i = -1\}$
- The payment is
$$f(x_1, \ldots, x_G) = \mu T^{G-C} \mathbf{1}\{W = 0\}.$$

---

The proposed mechanism has a *multiplicative* form: each answer in the gold standard is given a score based on whether it was correct (score = $\frac{1}{T}$), incorrect (score = 0) or skipped (score = 1), and the final payment is simply a product of these scores (scaled by $\mu$). The mechanism is easy to describe to workers: For instance, if $T = \frac{1}{2}$, $G = 3$ and $\mu = 80$ cents, then the description reads:

> *"The reward starts at 10 cents. For every correct answer in the 3 gold standard questions, the reward will double. However, if any of these questions are answered incorrectly, then the reward will become zero. So please use the 'I'm not sure' option wisely."*

Observe how this payment rule is similar to the popular 'double or nothing' paradigm [Dou14].

The algorithm makes a zero payment if *one or more* attempted answers in the gold standard are wrong. Note that this property is significantly stronger than the property of no-free-lunch which we originally required, where we wanted a zero payment only when *all* attempted answers were wrong. Surprisingly, as we prove shortly, Algorithm 1 is the only incentive-compatible mechanism that satisfies no-free-lunch.

The following theorem shows that the proposed payment mechanism indeed incentivizes a worker to skip the questions for which her confidence is below $T$, while answering those for which her confidence is greater than $T$. In the latter case, the worker is incentivized to select the answer which she thinks is most likely to be correct.

**Theorem 1.** *The payment mechanism of Algorithm 1 is incentive-compatible and satisfies the no-free-lunch condition.*

The proof of Theorem 1 is presented in Appendix A. It is easy to see that the mechanism satisfies no-free-lunch. The proof of incentive compatibility is also not hard: We consider any arbitrary worker (with arbitrary belief distributions), and compute the expected payment for that worker for the case when her choices in the task follow the requirements. We then show that any other choice leads to a strictly smaller expected payment.

While we started out with a very weak condition of no-free-lunch of making a zero payment when *all* attempted answers are wrong, the mechanism proposed in Algorithm 1 is significantly more strict and makes a zero payment when *any* of the attempted answers is wrong. A natural question that arises is: can we design an alternative mechanism satisfying incentive compatibility and no-free-lunch that operates somewhere in between?

## 3.2 Uniqueness of the Mechanism

In the previous section we showed that our proposed multiplicative mechanism is incentive compatible and satisfies the intuitive requirement of no-free-lunch. It turns out, perhaps surprisingly, that this mechanism is unique in this respect.

**Theorem 2.** *The payment mechanism of Algorithm 1 is the only incentive-compatible mechanism that satisfies the no-free-lunch condition.*

Theorem 2 gives a strong result despite imposing very weak requirements. To see this, recall our earlier discussion on deterring spammers, that is, incurring a low expenditure on workers who answer randomly. For instance, when the task comprises binary-choice questions, one may wish to design mechanisms which make a zero payment when the responses to $50\%$ or more of the questions in the gold standard are incorrect. The no-free-lunch axiom is a much weaker requirement, and the only mechanism that can satisfy this requirement is the mechanism of Algorithm 1.

The proof of Theorem 2 is available in Appendix B. The proof relies on the following key lemma that establishes a condition that any incentive-compatible mechanism must necessarily satisfy. The lemma applies to any incentive-compatible mechanism and not just to those satisfying no-free-lunch.

**Lemma.** *Any incentive-compatible payment mechanism $f$ must satisfy, for every $i \in \{1, \ldots, G\}$ and every $(y_1, \ldots, y_{i-1}, y_{i+1}, \ldots, y_G) \in \{-1, 0, 1\}^{G-1}$,*

$$Tf(y_1, \ldots, y_{i-1}, 1, y_{i+1}, \ldots, y_G) + (1 - T)f(y_1, \ldots, y_{i-1}, -1, y_{i+1}, \ldots, y_G)$$
$$= f(y_1, \ldots, y_{i-1}, 0, y_{i+1}, \ldots, y_G).$$

The proof of this lemma is provided in Appendix C. Given this lemma, the proof of Theorem 2 is then completed via an induction on the number of skipped questions.

## 3.3 Optimality against Spamming Behavior

As discussed earlier, crowdsouring tasks, especially those with multiple choice questions, often encounter spammers who answer randomly without heed to the question being asked. For instance, under a binary-choice setup, a spammer will choose one of the two options uniformly at random for every question. A highly desirable objective in crowdsourcing settings is to deter spammers. To this end, one may wish to impose a condition of zero payment when the responses to $50\%$ or more of the attempted questions in the gold standard are incorrect. A second desirable metric could be to minimize the expenditure on a worker who simply skips all questions. While the aforementioned requirements were deterministic functions of the worker's responses, one may alternatively wish to impose requirements that depend on the distribution of the worker's answering process. For instance, a third desirable feature would be to minimize the expected payment to a worker who answers all questions uniformly at random. We now show that interestingly, our unique multiplicative payment mechanism *simultaneously* satisfies all these requirements. The result is stated assuming a multiple-choice setup, but extends trivially to non-multiple-choice settings.

**Theorem 3.A** (Distributional). *Consider any value $A \in \{0, \ldots, G\}$. Among all incentive-compatible mechanisms (that may or may not satisfy no-free-lunch), Algorithm 1 strictly minimizes the expenditure on a worker who skips some $A$ of the questions in the the gold standard, and chooses answers to the remaining $(G - A)$ questions uniformly at random.*

**Theorem 3.B** (Deterministic). *Consider any value $B \in (0, 1]$. Among all incentive-compatible mechanisms (that may or may not satisfy no-free-lunch), Algorithm 1 strictly minimizes the expenditure on a worker who gives incorrect answers to a fraction $B$ or more of the questions attempted in the gold standard.*

The proof of Theorem 3 is presented in Appendix D. We see from this result that the multiplicative payment mechanism of Algorithm 1 thus possesses very useful properties geared to deter spammers, while ensuring that a good worker will be paid a high enough amount.

To illustrate this point, let us compare the mechanism of Algorithm 1 with the popular additive class of payment mechanisms.

**Example 1.** *Consider the popular class of "additive" mechanisms, where the payments to a worker are added across the gold standard questions. This additive payment mechanism offers a reward of $\frac{\mu}{G}$ for every correct answer in the gold standard, $\frac{\mu T}{G}$ for every question skipped, and $0$ for every incorrect answer. Importantly, the final payment to the worker is the* sum *of the rewards across the $G$ gold standard questions. One can verify that this additive mechanism is incentive compatible. One can also see that that as guaranteed by our theory, this additive payment mechanism does not satisfy the no-free-lunch axiom.*

*Suppose each question involves choosing from two options. Let us compute the expenditure that these two mechanisms make under a spamming behavior of choosing the answer randomly to each question. Given the $50\%$ likelihood of each question being correct, on can compute that the additive mechanism makes a payment of $\frac{\mu}{2}$ in expectation. On the other hand, our mechanism pays an expected amount of only $\mu 2^{-G}$. The payment to spammers thus reduces exponentially with the number of gold standard questions under our mechanism, whereas it does not reduce at all in the additive mechanism.*

*Now, consider a different means of exploiting the mechanism(s) where the worker simply skips all questions. To this end, observe that if a worker skips all the questions then the additive payment mechanism will incur an expenditure of $\mu T$. On the other hand, the proposed payment mechanism of Algorithm 1 pays an exponentially smaller amount of $\mu T^G$ (recall that $T < 1$).*

# 4 Simulations and Experiments

In this section, we present synthetic simulations and real-world experiments to evaluate the effects of our setting and our mechanism on the final label quality.

## 4.1 Synthetic Simulations

We employ synthetic simulations to understand the effects of various kinds of labeling errors in crowdsourcing. We consider binary-choice questions in this set of simulations. Whenever a worker answers a question, her confidence for the correct answer is drawn from a distribution $\mathcal{P}$ independent of all else. We investigate the effects of the following five choices of the distribution $\mathcal{P}$:

- The uniform distribution on the support $[0.5, 1]$.
- A triangular distribution with lower end-point $0.2$, upper end-point $1$ and a mode of $0.6$.
- A beta distribution with parameter values $\alpha = 5$ and $\beta = 1$.
- The hammer-spammer distribution [KOS11], that is, uniform on the discrete set $\{0.5, 1\}$.
- A truncated Gaussian distribution: a truncation of $\mathcal{N}(0.75, 0.5)$ to the interval $[0, 1]$.

When a worker has a confidence $p$ (drawn from the distribution $\mathcal{P}$) and attempts the question, the probability of making an error equals $(1 - p)$.

We compare (a) the setting where workers attempt every question, with (b) the setting where workers skip questions for which their confidence is below a certain threshold $T$. In this set of simulations, we set $T = 0.75$. In either setting, we aggregate the labels obtained from the workers for each question via a majority vote on the two classes. Ties are broken by choosing one of the two options uniformly at random.

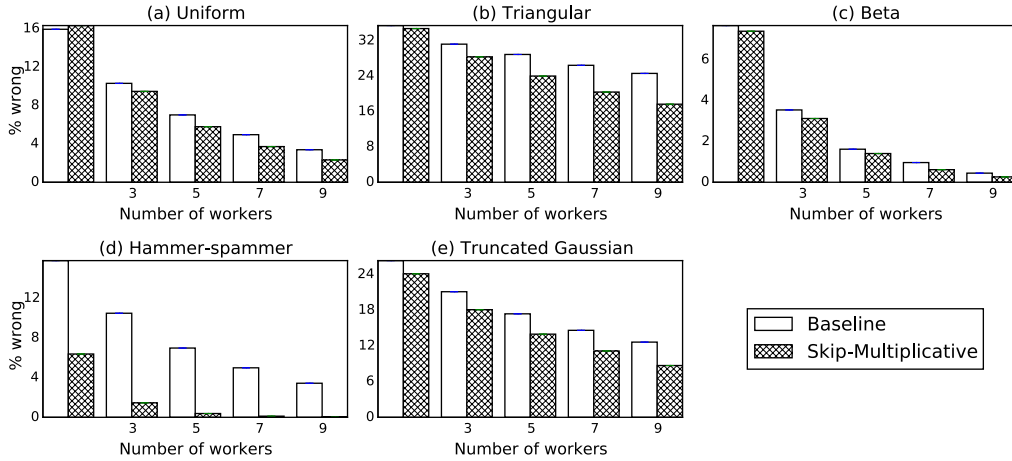

Figure 2: Error under different interfaces for synthetic simulations of five distributions of the workers' error probabilities.

Figure 2 depicts the results from these simulations. Each bar represents the fraction of questions that are labeled incorrectly, and is an average across 50,000 trials. (The standard error of the mean is too small to be visible.) We see that the skip-based setting consistently outperforms the conventional setting, and the gains obtained are moderate to high depending on the underlying distribution of the workers' errors. In particular, the gains are quite striking under the hammer-spammer model: this result is not surprising since the mechanism (ideally) screens the spammers out and leaves only the hammers who answer perfectly.

## 4.2 Experiments on Amazon Mechanical Turk

We conducted preliminary experiments on the Amazon Mechanical Turk commercial crowdsourcing platform (`mturk.com`) to evaluate our proposed scheme in real-world scenarios. The complete data, including the interface presented to the workers in each of the tasks, the results obtained from the workers, and the ground truth solutions, are available on the website of the first author.

**Goal.** Before delving into details, we first note certain caveats relating to such a study of mechanism design on crowdsourcing platforms. When a worker encounters a mechanism for only a small amount of time (a handful of tasks in typical research experiments) and for a small amount of money (at most a few dollars in typical crowdsourcing tasks), we cannot expect the worker to completely understand the mechanism and act precisely as required. For instance, we wouldn't expect our experimental results to change significantly even upon moderate modifications in the promised amounts, and furthermore, we do expect the outcomes to be noisy. Incentive compatibility kicks in when the worker encounters a mechanism across a longer term, for example, when a proposed mechanism is adopted as a standard for a platform, or when higher amounts are involved. This is when we would expect workers or others (e.g., bloggers or researchers) to design strategies that can game the mechanism. The theoretical guarantee of incentive compatibility or strict properness then prevents such gaming in the long run.

We thus regard these experiments as preliminary. Our intentions towards this experimental exercise were (a) to evaluate the potential of our algorithms to work in practice, and (b) to investigate the effect of the proposed algorithms on the net error in the collected labelled data.

**Experimental setup.** We conducted the five following experiments ("tasks") on Amazon Mechanical Turk: (a) identifying the golden gate bridge from pictures, (b) identifying the breeds of dogs from pictures, (c) identifying heads of countries, (d) identifying continents to which flags belong, and (e) identifying the textures in displayed images. Each of these tasks comprised 20 to 126 multi-

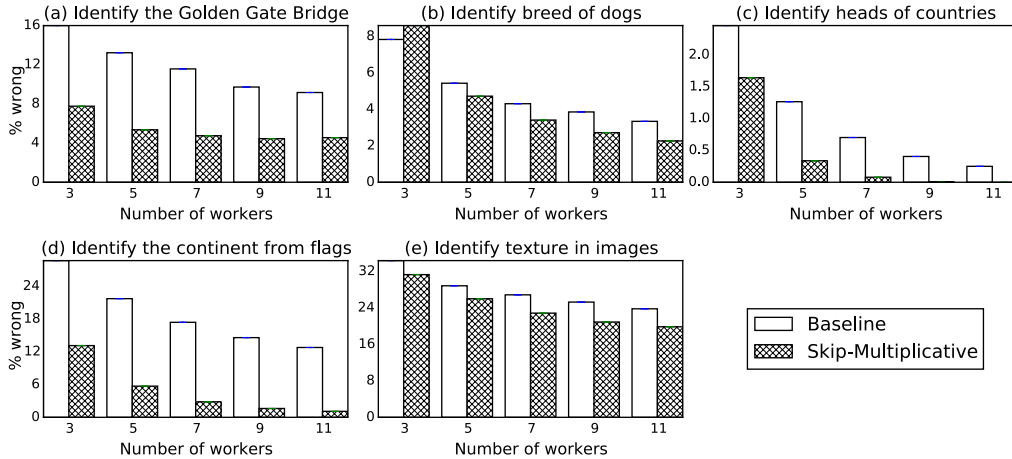

Figure 3: Error under different interfaces and mechanisms for five experiments conducted on Mechanical Turk.

ple choice questions.[3] For each experiment, we compared (i) a baseline setting (Figure 1a) with an additive payment mechanism that pays a fixed amount per correct answer, and (ii) our skip-based setting (Figure 1b) with the multiplicative mechanism of Algorithm 1. For each experiment, and for each of the two settings, we had 35 workers independently perform the task.

Upon completion of the tasks on Amazon Mechanical Turk, we aggregated the data in the following manner. For each mechanism in each experiment, we subsampled 3, 5, 7, 9 and 11 workers, and took a majority vote of their responses. We averaged the accuracy across all questions and across 1,000 iterations of this subsample-and-aggregate procedure.

**Results.** Figure 3 reports the error in the aggregate data in the five experiments. We see that in most cases, our skip-based setting results in a higher quality data, and in many of the instances, the reduction is two-fold or higher. All in all, in the experiments, we observed a substantial reduction in the amount of error in the labelled data while expending the same or lower amounts and receiving no negative comments from the workers. These observations suggest that our proposed skip-based setting coupled with our multiplicative payment mechanisms have potential to work in practice; the underlying fundamental theory ensures that the system cannot be gamed in the long run.

## 5   Discussion and Conclusions

In an extended version of this paper [SZ14], we generalize the "skip-based" setting considered here to one where we also elicit the workers' confidence about their answers. Moreover, in a companion paper [SZP15], we construct mechanisms to elicit the support of worker's beliefs.

Our mechanism offers some additional benefits. The pattern of skips of the workers provide a reasonable estimate of the difficulty of each question. In practice, the questions that are estimated to be more difficult may now be delegated to an expert or to additional non-expert workers. Secondly, the theoretical guarantees of our mechanism may allow for better post-processing of the data, incorporating the confidence information and improving the overall accuracy. Developing statistical aggregation algorithms or augmenting existing ones (e.g., [RYZ+10, KOS11, LPI12, ZLP+15]) for this purpose is a useful direction of research. Thirdly, the simplicity of our mechanisms may facilitate an easier adoption among the workers. In conclusion, given the uniqueness and optimality in theory, simplicity, and good performance observed in practice, we envisage our *multiplicative* payment mechanisms to be of interest to practitioners as well as researchers who employ crowdsourcing.

## Footnotes

[1] In the event that the confidence about a question is exactly equal to $T$, the worker may be equally incentivized to answer or skip.

[2]Such a payment function that is based on gold standard questions is also called a "strictly proper scoring rule" [GR07].

[3]See the extended version of this paper [SZ14] for additional experiments involving free-form responses, such as text transcription.

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
