[Supplementary Material]

# Supplementary material for "Double or Nothing: Multiplicative Incentive Mechanisms for Crowdsourcing"
## Nihar B. Shah and Dengyong Zhou

In this section, we provide proofs of the theoretical results presented in the paper.

## A    Proof of Theorem 1: Algorithm 1 is Incentive Compatible

The proposed payment mechanism satisfies the no-free-lunch condition since the payment is zero when there are one or more wrong answers. It remains to show that the mechanism is incentive compatible.

We will first assume that, for every question that the worker does not skip, she selects the answer which she believes is most likely to be correct. Under this assumption we will show that the worker is incentivized to skip the questions for which her confidence is smaller than $T$ and attempt if it is greater than $T$. Finally, we will show that the mechanism indeed incentivizes the worker to select the answer which she believes is most likely to be correct for the questions that she doesn't skip. In what follows, we will employ the notation $\kappa = \mu T^G$.

Let us first consider the case when $G = N$. Let $p_1, \ldots, p_N$ be the confidences of the worker for to questions $1, \ldots, N$ respectively. Further, let $p_{(1)} \geq \cdots \geq p_{(m)} > T > p_{(m+1)} \geq \cdots \geq p_{(N)}$ be the ordered permutation of these confidences (for some number $m$). Let $\{(1), \ldots, (N)\}$ denote the corresponding permutation of the $N$ questions. If the mechanism is incentive compatible, then the expected payment received by this worker should be maximized when the worker answers questions $(1), \ldots, (m)$ and skips the rest. Under the mechanism proposed in Algorithm 1, this action fetches the worker an expected payment of

$$\kappa \frac{p_{(1)}}{T} \cdots \frac{p_{(m)}}{T} \ .$$

Alternatively, if the worker answers the questions $\{i_1, \ldots, i_z\}$, with $p_{i_1} < \cdots < p_{i_y} < T < p_{i_{y+1}} < \cdots p_{i_z}$, then the expected payment is

$$p_{i_1} \cdots p_{i_z} \frac{\kappa}{T^z} \quad = \quad \kappa \frac{p_{i_1}}{T} \cdots \frac{p_{i_z}}{T} \tag{1}$$

$$\leq \quad \kappa \frac{p_{i_1}}{T} \cdots \frac{p_{i_y}}{T} \tag{2}$$

$$\leq \quad \kappa \frac{p_{(1)}}{T} \cdots \frac{p_{(m)}}{T} \tag{3}$$

where (2) is because $\frac{p_{i_j}}{T} \leq 1 \ \forall j > y$ and holds with equality only when $z = y$. Inequality (3) is a result of $\frac{p_{(j)}}{T} \geq 1 \ \forall j \leq m$ and holds with equality only when $y = m$. It follows that the expected payment is (strictly) maximized when $i_1 = (1), \ldots, i_z = (m)$ as required.

The case of $G < N$ is a direct consequence of the result for $G = N$, as follows. When $G < N$, from a worker's point of view, the set of $G$ questions is distributed uniformly at random in the superset of $N$ questions. However, for every set of $G$ questions, the relations (1), (2), (3) and their associated equality/strict-inequality conditions hold. The expected payment is thus (strictly) maximized when the worker answers the questions for which her confidence is greater than $T$ and skips those for which her confidence is smaller than $T$.

One can see that for every question that the worker chooses to answer, the expected payment increases with an increase in her confidence. Thus, the worker is incentivized to select the answer that she thinks is most probably correct.

Finally, since $\kappa = \mu T^G > 0$ and $T \in (0, 1)$, the payment is always non-negative and satisfies the $\mu$-budget constraint.

## B    Proof of Theorem 2: Uniqueness of Algorithm 1 under No-free-lunch

The proof is based on the following key lemma, establishing a condition that any incentive-compatible mechanism must necessarily satisfy. Note that this lemma does *not* require the no-free-lunch condition.

**Lemma 4.** *Any incentive-compatible mechanism $f$ must satisfy, for every $i \in \{1, \ldots, G\}$ and every* $(y_1, \ldots, y_{i-1}, y_{i+1}, \ldots, y_G) \in \{-1, 0, 1\}^{G-1}$,

$$Tf(y_1, \ldots, y_{i-1}, 1, y_{i+1}, \ldots, y_G) + (1 - T)f(y_1, \ldots, y_{i-1}, -1, y_{i+1}, \ldots, y_G)$$
$$= f(y_1, \ldots, y_{i-1}, 0, y_{i+1}, \ldots, y_G) .$$

The proof of Lemma 4 is provided in Appendix C.

We will first prove that any incentive-compatible mechanism satisfying the no-free-lunch condition must make a zero payment if one or more answers in the gold standard are incorrect. The proof proceeds by induction on the number of skipped questions $S$ in the gold standard. Let us assume for now that in the $G$ questions in the gold standard, the first question is answered incorrectly, the next $(G - 1 - S)$ questions are answered by the worker and have arbitrary evaluations, and the remaining $S$ questions are skipped. The proof proceeds by an induction on $S$. Suppose $S = G - 1$. In this case, the only attempted question is the first question and the answer provided by the worker to this question is incorrect. The no-free-lunch condition necessitates a zero payment in this case, thus satisfying the base case of our induction hypothesis. Now we prove the hypothesis for some $S$ under the assumption of it being true when the number of questions skipped in the gold standard is $(S + 1)$ or more. From Lemma 4 (with $i = G - S - 1$) we have

$$Tf(-1, y_2, \ldots, y_{G-S-2}, 1, 0, \ldots, 0) + (1 - T)f(-1, y_2, \ldots, y_{G-S-2}, -1, 0, \ldots, 0)$$
$$= f(-1, y_2, \ldots, y_{G-S-2}, 0, 0, \ldots, 0)$$
$$= 0,$$

where the final equation is a consequence of our induction hypothesis: The induction hypothesis is applicable since $f(-1, y_2, \ldots, y_{G-S-2}, 0, 0, \ldots, 0)$ corresponds to the case when the last $(S + 1)$ questions are skipped and the first question is answered incorrectly. Now, since the payment $f$ must be non-negative and since $T \in (0, 1)$, it must be that

$$f(-1, y_2, \ldots, y_{G-S-2}, 1, 0, \ldots, 0) = 0,$$

and

$$f(-1, y_2, \ldots, y_{G-S-2}, -1, 0, \ldots, 0) = 0.$$

This completes the proof of our induction hypothesis. Furthermore, each of the arguments above hold for any permutation of the $G$ questions, thus proving the necessity of zero payment when any one or more answers are incorrect.

We will now prove that when no answers in the gold standard are incorrect, the payment must be of the form described in Algorithm 1. Let $\kappa$ be the payment when all $G$ questions in the gold standard are skipped. Let $C$ be the number questions answered correctly in the gold standard. Since there are no incorrect answers, it follows that the remaining $(G - C)$ questions are skipped. Let us assume for now that the first $C$ questions are answered correctly and the remaining $(G - C)$ questions are skipped. We repeatedly apply Lemma 4, and the fact that the payment must be zero when one or more answers are wrong, to get

$$f(\underbrace{1, \ldots, 1}_{C-1}, 1, \underbrace{0, \ldots, 0}_{G-C}) = \frac{1}{T} f(\underbrace{1, \ldots, 1}_{C-1}, 0, \underbrace{0, \ldots, 0}_{G-C}) - \frac{1-T}{T} f(\underbrace{1, \ldots, 1}_{C-1}, -1, \underbrace{0, \ldots, 0}_{G-C})$$

$$= \frac{1}{T} f(\underbrace{1, \ldots, 1}_{C-1}, 0, \underbrace{0, \ldots, 0}_{G-C})$$

$$\vdots$$

$$= \frac{1}{T^C} f(\underbrace{0, \ldots, 0}_{G})$$

$$= \frac{1}{T^C} \kappa .$$

In order to abide by the budget, we must have the maximum payment as $\mu = \kappa \frac{1}{T^G}$. It follows that $\kappa = \mu T^G$. Finally, the arguments above hold for any permutation of the $G$ questions, thus proving the uniqueness of the mechanism of Algorithm 1.

## C  Proof of Lemma 4: Necessary Condition for any Incentive-compatible Mechanism

First we consider the case of $G = N$. In the set $\{y_1, \ldots, y_{i-1}, y_{i+1}, \ldots, y_G\}$, for some $(\eta, \gamma) \in \{0, \ldots, G-1\}^2$, suppose there are $\eta$ elements with a value 1, $\gamma$ elements with a value $-1$, and $(G - 1 - \eta - \gamma)$ elements with a value 0. Let us assume for now that $i = \eta + \gamma + 1$, $y_1 = 1, \ldots, y_\eta = 1, y_{\eta+1} = -1, \ldots, y_{\eta+\gamma} = -1, y_{\eta+\gamma+2} = 0, \ldots, y_G = 0$.

Suppose the worker has confidences $(p_1, \ldots, p_{\eta+\gamma}) \in (T, 1]^{\eta+\gamma}$ for the first $(\eta + \gamma)$ questions, a confidence of $q \in (0, 1]$ for the next question, and confidences smaller than $T$ for the remaining $(G - \eta - \gamma - 1)$ questions. The mechanism must incentivize the worker to answer the first $(\eta + \gamma)$ questions and skip the last $(G - \eta - \gamma - 1)$ questions; for question $(\eta + \gamma + 1)$, it must incentivize the worker to answer if $q > T$ and skip if $q < T$. Supposing the worker indeed attempts the first $(\eta + \gamma)$ questions and skips the last $(G - \eta - \gamma - 1)$ questions, let $\boldsymbol{x} = \{x_1, \ldots, x_{\eta+\gamma}\} \in \{-1, 1\}^{\eta+\gamma}$ denote the the evaluation of the worker's answers to the first $(\eta + \gamma)$ questions. Define quantities $\{r_j\}_{j \in [\eta+\gamma]}$ as $r_j = 1 - p_j$ for $j \in \{1, \ldots, \eta\}$, and $r_j = p_j$ for $j \in \{\eta+1, \eta+\gamma\}$. The requirement of incentive compatibility necessitates

$$q \sum_{\boldsymbol{x} \in \{-1,1\}^{\eta+\gamma}} \left( f(x_1, \ldots, x_\eta, -x_{\eta+1}, \ldots, -x_{\eta+\gamma}, 1, 0, \ldots, 0) \prod_{j \in [\eta+\gamma]} r_j^{\frac{1-x_j}{2}} (1 - r_j)^{\frac{1+x_j}{2}} \right)$$

$$+ (1-q) \sum_{\boldsymbol{x} \in \{-1,1\}^{\eta+\gamma}} \left( f(x_1, \ldots, x_\eta, -x_{\eta+1}, \ldots, -x_{\eta+\gamma}, -1, 0, \ldots, 0) \prod_{j \in [\eta+\gamma]} r_j^{\frac{1-x_j}{2}} (1 - r_j)^{\frac{1+x_j}{2}} \right)$$

$$\overset{q<T}{\underset{q>T}{\lessgtr}} \sum_{\boldsymbol{x} \in \{-1,1\}^{\eta+\gamma}} \left( f(x_1, \ldots, x_\eta, -x_{\eta+1}, \ldots, -x_{\eta+\gamma}, 0, 0, \ldots, 0) \prod_{j \in [\eta+\gamma]} r_j^{\frac{1-x_j}{2}} (1 - r_j)^{\frac{1+x_j}{2}} \right).$$

The left hand side of this expression is the expected payment if the worker chooses to answer question $(\eta + \gamma + 1)$, while the right hand side is the expected payment if she chooses to skip it. For any real-valued variable $q$, and for any real-valued constants $a$, $b$ and $c$,

$$aq \overset{q<c}{\underset{q>c}{\lessgtr}} b \quad \Rightarrow \quad ac = b.$$

As a result,

$$T \sum_{\boldsymbol{x} \in \{-1,1\}^{\eta+\gamma}} \left( f(x_1, \ldots, x_\eta, -x_{\eta+1}, \ldots, -x_{\eta+\gamma}, 1, 0, \ldots, 0) \prod_{j \in [\eta+\gamma]} r_j^{\frac{1-x_j}{2}} (1 - r_j)^{\frac{1+x_j}{2}} \right)$$

$$+ (1-T) \sum_{\boldsymbol{x} \in \{-1,1\}^{\eta+\gamma}} \left( f(x_1, \ldots, x_\eta, -x_{\eta+1}, \ldots, -x_{\eta+\gamma}, -1, 0, \ldots, 0) \prod_{j \in [\eta+\gamma]} r_j^{\frac{1-x_j}{2}} (1 - r_j)^{\frac{1+x_j}{2}} \right)$$

$$- \sum_{\boldsymbol{x} \in \{-1,1\}^{\eta+\gamma}} \left( f(x_1, \ldots, x_\eta, -x_{\eta+1}, \ldots, -x_{\eta+\gamma-1}, 0, 0, \ldots, 0) \prod_{j \in [\eta+\gamma]} r_j^{\frac{1-x_j}{2}} (1 - r_j)^{\frac{1+x_j}{2}} \right) = 0. \tag{4}$$

The left hand side of (4) represents a polynomial in $(\eta + \gamma)$ variables $\{r_j\}_{j=1}^{\eta+\gamma}$ which evaluates to zero for all values of the variables within a $(\eta + \gamma)$-dimensional solid Euclidean ball. Thus, the coefficients of the monomials in this polynomial must be zero. In particular, the constant term must be zero. The constant term appears when $x_j = 1 \, \forall \, j$ in the summations in (4). Setting the constant term to zero gives

$$Tf(x_1 = 1, \ldots, x_\eta = 1, -x_{\eta+1} = -1, \ldots, -x_{\eta+\gamma} = -1, 1, 0, \ldots, 0)$$
$$+ (1 - T)f(x_1 = 1, \ldots, x_\eta = 1, -x_{\eta+1} = -1, \ldots, -x_{\eta+\gamma} = -1, -1, 0, \ldots, 0)$$
$$- f(x_1 = 1, \ldots, x_\eta = 1, -x_{\eta+1} = -1, \ldots, -x_{\eta+\gamma} = -1, 0, 0, \ldots, 0) = 0$$

as desired. Since the arguments above hold for any permutation of the $G$ questions, this completes the proof for the case of $G = N$.

Now consider the case $G < N$. Let $g : \{-1, 0, 1\}^N \to \mathbb{R}_+$ represent the expected payment given an evaluation of all the $N$ answers, when the identities of the gold standard questions are unknown. Here, the expectation is with respect to the (uniformly random) choice of the $G$ gold standard questions. If $(x_1, \ldots, x_N) \in \{-1, 0, 1\}^N$ are the evaluations of the worker's answers to the $N$ questions then the expected payment is

$$g(x_1, \ldots, x_N) = \frac{1}{\binom{N}{G}} \sum_{(i_1, \ldots, i_G) \subseteq \{1, \ldots, N\}} f(x_{i_1}, \ldots, x_{i_G}) \,. \tag{5}$$

Notice that when $G = N$, the functions $f$ and $g$ are identical.

In the set $\{y_1, \ldots, y_{i-1}, y_{i+1}, \ldots, y_G\}$, for some $(\eta, \gamma) \in \{0, \ldots, G-1\}^2$ with $\eta + \gamma < G$, suppose there are $\eta$ elements with a value 1, $\gamma$ elements with a value $-1$, and $(G - 1 - \eta - \gamma)$ elements with a value 0. Let us assume for now that $i = \eta + \gamma + 1$, $y_1 = 1, \ldots, y_\eta = 1, y_{\eta+1} = -1, \ldots, y_{\eta+\gamma} = -1, y_{\eta+\gamma+2} = 0, \ldots, y_G = 0$.

Suppose the worker has confidences $\{p_1, \ldots, p_{\eta+\gamma}\} \in (T, 1]^{\eta+\gamma}$ for the first $(\eta + \gamma)$ of the $N$ questions, a confidence of $q \in (0, 1]$ for the next question, and confidences smaller than $T$ for the remaining $(N - \eta - \gamma - 1)$ questions. The mechanism must incentivize the worker to answer the first $(\eta + \gamma)$ questions and skip the last $(N - \eta - \gamma - 1)$ questions; for the $(\eta + \gamma + 1)^{\text{th}}$ question, the mechanism must incentivize the worker to answer if $q > T$ and skip if $q < T$. Supposing the worker indeed attempts the first $(\eta + \gamma)$ questions and skips the last $(N - \eta - \gamma - 1)$ questions, let $\boldsymbol{x} = \{x_1, \ldots, x_{\eta+\gamma}\} \in \{-1, 1\}^{\eta+\gamma}$ denote the the evaluation of the worker's answers to the first $(\eta + \gamma)$ questions. Define quantities $\{r_j\}_{j \in [\eta+\gamma]}$ as $r_j = 1 - p_j$ for $j \in \{1, \ldots, \eta\}$, and $r_j = p_j$ for $j \in \{\eta + 1, \eta + \gamma\}$. The requirement of incentive compatibility necessitates

$$q \sum_{\boldsymbol{x} \in \{-1,1\}^{\eta+\gamma}} \left( g(x_1, \ldots, x_\eta, -x_{\eta+1}, \ldots, -x_{\eta+\gamma}, 1, 0, \ldots, 0) \prod_{j \in [\eta+\gamma]} r_j^{\frac{1-x_j}{2}} (1 - r_j)^{\frac{1+x_j}{2}} \right)$$

$$+ (1-q) \sum_{\boldsymbol{x} \in \{-1,1\}^{\eta+\gamma}} \left( g(x_1, \ldots, x_\eta, -x_{\eta+1}, \ldots, -x_{\eta+\gamma}, -1, 0, \ldots, 0) \prod_{j \in [\eta+\gamma]} r_j^{\frac{1-x_j}{2}} (1 - r_j)^{\frac{1+x_j}{2}} \right)$$

$$\overset{q<T}{\underset{q>T}{\lessgtr}} \sum_{\boldsymbol{x} \in \{-1,1\}^{\eta+\gamma}} \left( g(x_1, \ldots, x_\eta, -x_{\eta+1}, \ldots, -x_{\eta+\gamma}, 0, 0, \ldots, 0) \prod_{j \in [\eta+\gamma]} r_j^{\frac{1-x_j}{2}} (1 - r_j)^{\frac{1+x_j}{2}} \right) \,. \tag{6}$$

Again, applying the fact that for any real-valued variable $q$ and for any real-valued constants $a$, $b$ and $c$, $aq \overset{q<c}{\underset{q>c}{\lessgtr}} b \implies ac = b$, we get that

$$T g(x_1 = 1, \ldots, x_\eta = 1, -x_{\eta+1} = -1, \ldots, -x_{\eta+\gamma} = -1, 1, 0, \ldots, 0)$$
$$+ (1 - T) g(x_1 = 1, \ldots, x_\eta = 1, -x_{\eta+1} = -1, \ldots, -x_{\eta+\gamma} = -1, -1, 0, \ldots, 0)$$
$$- g(x_1 = 1, \ldots, x_\eta = 1, -x_{\eta+1} = -1, \ldots, -x_{\eta+\gamma} = -1, 0, 0, \ldots, 0) = 0 \,. \tag{7}$$

The proof now proceeds via induction on the quantity $(G - \eta - \gamma - 1)$, i.e., on the number of skipped questions in $\{y_1, \ldots, y_{i-1}, y_{i+1}, \ldots, y_G\}$. We begin with the case of $(G - \eta - \gamma - 1) = G - 1$ which implies $\eta = \gamma = 0$. In this case (7) simplifies to

$$T g(1, 0, \ldots, 0) + (1 - T) g(-1, 0, \ldots, 0) = g(0, 0, \ldots, 0) \,.$$

Applying the expansion of function $g$ in terms of function $f$ from (5) gives

$$T \left( c_1 f(1, 0, \ldots, 0) + c_2 f(0, 0, \ldots, 0) \right) + (1 - T) \left( c_1 f(-1, 0, \ldots, 0) + c_2 f(0, 0, \ldots, 0) \right)$$
$$= \left( c_1 f(0, 0, \ldots, 0) + c_2 f(0, 0, \ldots, 0) \right)$$

for constants $c_1 > 0$ and $c_2 > 0$ that respectively denote the probabilities that the first question is picked and not picked in the set of $G$ gold standard questions. Cancelling out the common terms on both sides of the equation, we get the desired result

$$T f(1, 0, \ldots, 0) + (1 - T) f(-1, 0, \ldots, 0) = f(0, 0, \ldots, 0) \,.$$

Next, we consider the case when $(G - \eta - \gamma - 1)$ questions are skipped in the gold standard, and assume that the result is true when more than $(G - \eta - \gamma - 1)$ questions are skipped in the gold standard. In (7), the functions $g$ decompose into a sum of the constituent $f$ functions. These constituent functions $f$ are of two types: the first where all of the first $(\eta + \gamma + 1)$ questions are included in the gold standard, and the second where one or more of the first $(\eta + \gamma + 1)$ questions are not included in the gold standard. The second case corresponds to situations where there are more than $(G - \eta - \gamma - 1)$ questions skipped in the gold standard and hence satisfies our induction hypothesis. The terms corresponding to these functions thus cancel out in the expansion of (7). The remainder comprises only evaluations of function $f$ for arguments in which the first $(\eta + \gamma + 1)$ questions are included in the gold standard: since the last $(N - \eta - \gamma - 1)$ questions are skipped by the worker, the remainder evaluates to

$$Tc_3 f(y_1, \ldots, y_{\eta+\gamma}, 1, 0, \ldots, 0) + (1 - T)c_3 f(y_1, \ldots, y_{\eta+\gamma}, -1, 0, \ldots, 0)$$
$$= c_3 f(y_1, \ldots, y_{\eta+\gamma}, 0, 0, \ldots, 0)$$

for some constant $c_3 > 0$. Dividing throughout by $c_3$ gives the desired result.

Finally, the arguments above hold for any permutation of the first $G$ questions, thus completing the proof.

## D    Proof of Theorem 3: Minimum Payment to Spammers

**Part A (Distributional).** Let $m$ denote the number of options in each question. One can verify that under the mechanism of Algorithm 1, a worker who skips $A$ questions and answers the rest uniformly at random will get a payment of $\frac{\mu T^A}{m^{G-A}}$ in expectation. This expression arises due to the fact that Algorithm 1 makes a zero payment if any of the attempted answers are incorrect, and a payment of $\mu T^A$ if the worker skips $A$ questions and answers the rest correctly. Under uniformly random answers, the probability of the latter event is $\frac{1}{m^{G-A}}$.

Now consider any other mechanism, and denote it as $f'$. Let us suppose without loss of generality that the worker attempts the first $(G - A)$ questions. Since the payment must be non-negative, a repeated application of Lemma 4 gives

$$f'(\underbrace{1, \ldots, 1}_{G-A}, 0, \ldots, 0) \geq Tf'(\underbrace{1, \ldots, 1}_{G-A+1}, 0, \ldots, 0) \tag{8}$$

$$\vdots$$

$$\geq T^A f'(1, \ldots, 1)$$
$$= T^A \mu, \tag{9}$$

where (9) is a result of the $\mu$-budget constraint. Since there is a $\frac{1}{m^{G-A}}$ chance of the $(G - A)$ attempted answers being correct, the expected payment under any other mechanism $f'$ must be at least $\frac{\mu T^A}{m^{G-A}}$.

We will now show that if any mechanism $f'$ achieves the bound (9) with equality, then the mechanism must be identical to Algorithm 1. We split the proof of this part into two cases, depending on the value of the parameter $A$.

Case I $(A < G)$: In order to achieve the bound (9) with equality, the mechanism must make a zero payment if any of the $(G - A)$ attempted questions are answered incorrectly, that is, it must satisfy

$$f'(y_1, \ldots, y_{G-A}, 0, \ldots, 0) = 0 \qquad \forall (y_1, \ldots, y_{G-A}) \in \{-1, 1\}^{G-A} \backslash \{1\}^{G-A}.$$

A repeated application of Lemma 4 then implies

$$f'(-1, 0, \ldots, 0) = 0. \tag{10}$$

Note that so far we considered the case when the worker attempts the first $(G - A)$ questions. The arguments above hold for any choice of the $(G - A)$ attempted questions, and consequently the results shown so far in this proof hold for all permutations of the arguments to $f'$. In particular, the mechanism $f'$ must make a zero payment when any $(G - 1)$ questions in the gold standard

are skipped and the remaining question is answered incorrectly. Another repeated application of Lemma 4 to this result gives

$$f'(y_1, \ldots, y_G) = 0 \qquad \forall (y_1, \ldots, y_G) \in \{0, -1\}^G \backslash \{0\}^G.$$

This condition is precisely the no-free-lunch axiom, and in Theorem 2 we had shown that Algorithm 1 is the only incentive-compatible mechanism that satisfies this axiom. It follows that our mechanism, Algorithm 1 strictly minimizes the expected payment in the setting under consideration.

Case II ($A = G$): In order to achieve the bound (9) with equality, the mechanism $f'$ must also achieve the bound (8) with equality. Noting that we have $A = G$ in this case, it follows that the mechanism $f'$ must satisfy

$$f'(-1, 0, \ldots, 0) = 0.$$

This condition is identical to (10) established for Case I earlier, and the rest of the argument now proceeds in a manner identical to the subsequent arguments in Case I.

**Part B (Deterministic).** Algorithm 1 makes a payment of zero when one or more of the answers to questions in the gold standard are incorrect. Consequently, for every value of parameter $B \in (0, 1]$, Algorithm 1 makes a zero payment when a fraction $B$ or more of the attempted answers are incorrect. Any other mechanism doing so must satisfy the no-free-lunch axiom. In Theorem 2 we had shown that Algorithm 1 is the only incentive-compatible mechanism that satisfies this axiom. It follows that our mechanism, Algorithm 1 strictly minimizes the payment in the event under consideration.