[Reviews · NeurIPS 2015]

Submitted by Assigned_Reviewer_1

The paper proposes a multiplicative payment scheme for crowdsourcing workers. The proposed scheme is optimal according to different criteria, and it allows for minimizing the amount paid to spammers, while also motivating legitimate workers to improve the quality of their answers. The paper is well written, and theoretically and experimentally sound.

I have only some minor comments that may be worth addressing:

1) This approach can deal with spammers, as it aims to minimize the amount paid to them. How about trying to detect them and give them a zero reward, instead of a 'minimum' amount of money? Would that be more convenient? My impression is that, under the current model of a spammer (i.e., a person randomly selecting among the possible answers), detecting spammers explicitly may be easy. One very recent and related work on the detection of malicious crowdsourcing campaigns (called crowdturfing) is: -- G. Wang, T. Wang, H. Zheng, and B. Y. Zhao. Man vs. machine: Practical adversarial detection of malicious crowdsourcing workers. In 23rd USENIX Security Symposium (USENIX Security 14), San Diego, CA, 2014. USENIX Association. This point may deserve discussion in the paper.

2) Another observation (for potential future work) may be related to modeling the interaction between spammers and payers as a (repeated) game and study whether an equilibrium may exist (and if it is unique, depending on the kind of game modeled). This may be interesting since, as the authors also mentioned, the interactions between spammers and payers can be seen as a long-lasting arms race, or game, indeed. And spammers are expected to evolve from the 'random' model if they know that the incentive/payment mechanism has changed. Furthermore, modeling such interactions in a sound theoretical way may also provide additional guidelines on how to select the number of gold questions and the overall number of questions for each task. This is something that also deserves more discussion in the paper.

3) Another point that deserves some clarification/discussion is related to the assumption that the worker's confidences for different questions are independent. As this may be probably easily verified for a spammer that answers at random, I'm a bit more skeptical that this holds for legitimate workers. Given the experimental analysis and the available data, the authors may potentially verify whether this assumption holds in practice, or the extent to which it may be violated.

4) Finally, it would be also nice to put more emphasis in the experimental section of the paper on how the proposed payment mechanism distributes payments to workers while reducing the overall cost, and thus making the crowdsourcing campaign more effective. This is well-explained in the appendix, but not properly pointed out in the paper, in my opinion.

Overall, I found this work interesting, original, and theoretically sound. Experiments are also clear and convincing, and thus I can reasonably recommend acceptance of this work at NIPS.
Summary: The paper proposes a multiplicative payment scheme for crowdsourcing workers. It shows that the proposed scheme is optimal according to different criteria. In practice, it minimizes the amount paid to spammers, and motivates legitimate workers to improve the quality of their answers. The paper is well written, and theoretically and experimentally sound.

Submitted by Assigned_Reviewer_2

Paper summary: The paper proposes a payment rule for crowdsourced tasks. This rule is intended to incentivize workers to accurately report their confidence (e.g. by skipping a task when they have low confidence), and to pay little to spammers. Payment is based on the product of the evaluations of a worker's responses to a set of gold-standard tasks; if the worker gets a single gold standard task wrong and asserts high confidence, the overall payment is zero.

Quality: The theoretical developments seem sound and interesting, although I didn't check the proofs in the appendix.

My main concern is with the experiments. While the paper points out that the experiments are preliminary and that it's difficult to test incentive compatibility with short-term tasks, I'm not convinced that the experiments support the theoretical developments at all. There are two changes between baseline and experimental condition: first, the incentive rule changes; second, the user interface and data analysis method changes (allows skipping, confidence). Wouldn't a better test of the theoretical developments only change the incentive rule? That is, wouldn't it be better to have experiments that are exactly like experiments (ii) and (iii), except that users are shown another simple (presumably not incentive-compatible) payment rule?

As it stands, to me the most likely interpretation of the experimental results is that allowing users to skip tasks can increase data quality. I'm worried that readers will come away with the impression that the scoring rule was responsible for the drop in error rates when we don't really have evidence that that's the case. I understand that actually showing an effect of the incentive rule could require different experiments (possibly requiring the same worker to do many more tasks), but it does not seem infeasible. If it is infeasible, I'd still suggest doing the closer controls for (ii) and (iii) and being very clear about what part of the improvement in error beyond (i) is due to the change in setup, and what part is due to the change in scoring rule.

Clarity: The paper is clear and for the most part easy to read. It could be good to give the key insight behind some of the proofs in the main paper.

Originality: As far as I can tell, this is significantly different from prior work on payment rules for crowdsourcing. This would be easier to judge if the paper contained a more thorough/explicit discussion of related work. For example, are multiplicative payment rules used in other domains?

Significance: The development of incentive-compatible payment rules for crowdsourcing is highly relevant for the NIPS community and beyond, and this paper makes significant progress on the subset of this problem where gold standard answers are available.
Summary: This paper represents a significant advance in the theory of incentive compatible payment mechanisms for crowdsourcing. However, as it is, it does not present the most relevant experimental evidence, which (at the very least) needs to be clarified in the paper.

Submitted by Assigned_Reviewer_3

Authors consider a setting in which workers can either choose to answer or to "skip" the question. They define the "no free-lunch condition", which states that a payment should give reward zero if all answers (in the ground truth subset) are wrong.

The payment rule is very simple: either zero reward if one or more answers are wrong or a reward that grows (decreases for each skipped answer) multiplicatively. Theoretical results regarding the correctness and uniqueness of the proposed method are derived. The results are generalized to the case of different degrees of confidence and evaluated in a real experiment.

It is indeed surprising that the payment rule that has to give zero reward as soon as the worker answers incorrectly one question.

The key idea is very simple and the authors did a good job on the theoretical side. I have doubts, however, that NIPS should be the conference for this type of work. Nevertheless, I have checked in depth until page 14 in the supplementary material, which includes the proof of uniqueness (the most important). I would like to hear the opinion of the authors on the following points:

1- I can replace the factor $T^{G-C}$ by $T^{k(G-C)}$, for arbitrary $k$. This is a more general scoring rule that also satisfies the no-free lunch condition. For k=1, we recover the proposed rule. For k > 1 we penalize more the skips and or k < 1 (and here comes a problem) we penalize less the skips. I say this is a problem because in the limit of k->0 we do not penalize skipping, and the rule would give full reward for a all-skipped answer, something clearly not desired.

2- Concerning the empirical evaluation, there is significant selection bias in the experiments: only the good workers "having at least 95% of their prior work approved by the requesters" are selected for the experiment (lines 1115-1116 suppl). This seems to be a critical point because the main motivation is to defer spammers, and this is not really evaluated. Authors should mention that as an (additional) limitation in the beginning of section 5.

Related to this, if the main goal is to compare with the traditional method, wouldn't be worthy to make a simulation where the p's are generated randomly without the confounders of a real experiment?

3a- As authors note in lines 234-238, this payment rule is indeed restrictive. For example, given N questions, performing three different tasks is different than performing one task only, since failing one single question will lead to zero reward in the later and will not in the former. It is mentioned that in reality, "workers play a repeated game", but this is not considered at all in the analysis/evaluation.

3b- Finally, and related to the previous point, note that the proposed rule is more informative than the standard and can leak information, in the sense that zero reward means that one question of the gold standard was answered incorrectly. This could be used by a group of cheaters to learn the subset of gold standard questions.

I have other minor remarks:

- In Axiom 1, why not replacing the inequality and equality just with the condition $\sum_{i=1}^G\one\{x_i=-1\} = G$? - line 510 suppl: $kappa$ should be $\kappa$ - line 577 suppl: $j\in[\eta+\gamma]$ should be

$j\in[1,\eta+\gamma]$. This should also be corrected many times later on. - line 577 suppl: $j\in\{\eta+1,\eta+\gamma\}$ should be $j\in\{\eta+1,\hdots,\eta+\gamma\}$ - It is not clear why the wrong answers are changed (eg, lines 580-588 suppl). - line 738 suppl: Algorithm ?? - (lines 1199, 1209) suppl: the user is informed that the bonus is not altered if a question is skipped, but that is not true, since skipping decreases reward exponentially.
Summary: The paper considers the problem of low quality responses in crowdsourcing tasks and proposes a payment rule that has interesting theoretical properties. Overall, this is a good paper and authors did a good job. There are some issues regarding the uniqueness of the payment rule and on the experimental evaluation that need to be addressed. It might also be questionable if NIPS is a good place for this work.

Author Feedback
Author rebuttal: We thank all the reviewers for reading the paper and providing valuable feedback. The specific questions of each reviewer are addressed below.

Reviewer_1:

1) The reference is certainly of interest and nicely complements our work

2) If the order of options is shuffled for every question, or if the true answers are uniformly distributed on the options, then our results continue to hold for any adaptive strategy of a 'spammer' (one who answers without regard to the question being asked)

3) This is a simplifying assumption for analysis. While the popular notion of bounded rationality does lend credence to this assumption, we agree that a further exploration is useful. Note that this assumption is not needed for Theorems 2 and 5 which show that no other incentive-compatible mechanism can satisfy no-free-lunch.

Reviewer_2:

- The experiments presented in the paper show the improvements due to using the (skip interface+our mechanism) together. In a separate set of three experiments involving over 300 workers, we compared fixed payment mechanisms (which are the most commonly used mechanisms on Mechanical Turk) with incentive-compatible multiplicative mechanisms across the same interface. In each case, the difference in the data obtained from the two mechanisms was statistically significant (p-value < 0.01 in each experiment).

- We are unaware of literature on multiplicative rules, except for casinos and online/TV games that frequently employ such mechanisms ("double or nothing"). We do not know of any existing non-trivial theoretical guarantees associated to these rules.

Reviewer_3:

- The factor T^{G-C} cannot be replaced by T^{k(G-C)}: the resulting mechanism is not incentive compatible. It is incentive compatible only when k=1 as proposed in Algorithm 1 of our paper. (Proof of this follows from Theorem 2.) In general, exponentiating an incentive-compatible mechanism may not preserve incentive compatibility.

- Relevance to NIPS: Our paper is submitted under the primary category "game theory" in NIPS 2015. The use of crowdsourcing to obtain labeled data for machine learning algorithms is very popular in both academia (ImageNet) and the industry(Facebook/Google/Microsoft). Our work provides practitioners an efficient way to obtain high quality labels which are critical for the success of many machine learning applications in practice.

- It is common for requesters to use the standard Mechanical Turk filter to hire workers with 95% acceptance rate. However, we agree that it will be interesting to see the outcome otherwise. We conjecture that it will only further improve our mechanism's performance relative to the baseline, since our mechanism ensures a minimum payment to spammers and hence will spend lower amounts.

- We conducted synthetic simulations as suggested by the reviewer, for various settings. For every distribution of p's we evaluated, we observed moderate to high reduction in error as compared to the baseline. Here are the *reduction in the error* as compared to the baseline for aggregation across 5 workers with T=0.75 for various distributions: Beta(5,1): 16%, Uniform: 9%, Hammer-spammer [Karger-Oh-Shah, NIPS11]: 86%, Truncated Normal(.9,.5): 25%, Truncated Normal(.75,.5): 13%, Truncated Normal(.6,.5): 4%

- The notion of repeated game is used only to assert that the average hourly wage of the worker converges quickly to its expected value, allowing the assumption that workers optimize for their expected pay. Given this assumption and existence of gold standards, one can consider a single worker in a single task without loss of generality; our theoretical guarantees are retained even in a repeated game setting.

- Every incentive compatible mechanism must leak this information: Incentive comaptibility requires payment to be an injective function, and hence the payment will have to reveal the number of gold standard questions answered correctly. In commercial crowdsourcing platforms, there is little known evidence for collaborative cheating.

- (1199,1209 suppl): The instructions are correct, and simply paraphrase Algorithm 1 (for the skip-based setting): Bonus starts at $\mu T^G$, reduces to 0 for a wrong answer, is multiplied by 1/T for every correct answer, and remains unchanged for a skip. Likewise for confidence-based.

Reviewer_4:

- Every possible incentive-compatible mechanism must pay a certain amount of money to a worker who skips everything, i.e., a zero payment to workers who skip everything is non-incentive-compatible.

- Our mechanism is the only one that can pay the minimum amount of this necessary payment. As compared to the maximum payment, the payment under our mechanism is *exponentially small* in the case of all skips.

- Coupled with our mechanism, the option to skip can dramatically reduce label noise as compared to the typical setting where skips are not allowed.